# The ATR Inhibitor VE-821 Enhances the Radiosensitivity and Suppresses DNA Repair Mechanisms of Human Chondrosarcoma Cells

**DOI:** 10.3390/ijms24032315

**Published:** 2023-01-24

**Authors:** Birgit Lohberger, Dietmar Glänzer, Nicole Eck, Katharina Stasny, Anna Falkner, Andreas Leithner, Dietmar Georg

**Affiliations:** 1Department of Orthopedics and Trauma, Medical University of Graz, 8036 Graz, Austria; 2MedAustron Ion Therapy Center, 2700 Wiener Neustadt, Austria; 3Department of Radiation Oncology, Medical University of Vienna, 1090 Vienna, Austria

**Keywords:** chondrosarcoma, particle therapy, proton irradiation, carbon ion irradiation, DNA repair, VE-821

## Abstract

To overcome the resistance to radiotherapy in chondrosarcomas, the prevention of efficient DNA repair with an additional treatment was explored for particle beams as well as reference X-ray irradiation. The combined treatment with DNA repair inhibitors—with a focus on ATRi VE-821—and proton or carbon ions irradiation was investigated regarding cell viability, proliferation, cell cycle distribution, MAPK phosphorylation, and the expression of key DNA repair genes in two human chondrosarcoma cell lines. Pre-treatment with the PARPis Olaparib or Veliparib, the ATMi Ku-55933, and the ATRi VE-821 resulted in a dose-dependent reduction in viability, whereas VE-821 has the most efficient response. Quantification of γH2AX phosphorylation and protein expression of the DNA repair pathways showed a reduced regenerative capacity after irradiation. Furthermore, combined treatment with VE-821 and particle irradiation increased MAPK phosphorylation and the expression of apoptosis markers. At the gene expression and at the protein expression/phosphorylation level, we were able to demonstrate the preservation of DNA damage after combined treatment. The present data showed that the combined treatment with ATMi VE-821 increases the radiosensitivity of human chondrosarcoma cells in vitro and significantly suppresses efficient DNA repair mechanisms, thus improving the efficiency of radiotherapy.

## 1. Introduction

Chondrosarcoma represents a heterogeneous group of locally aggressive and malignant entities and is the second most common primary malignant bone tumor after osteosarcoma. Overall survival and prognosis depend on histological grade and tumor subtype [1]. Poor vascularization, a slow division rate, and a hyaline cartilage matrix that prevents access to the cells are reasons for the existing resistance to chemotherapy and radiotherapy. For this reason, therapy options are limited and complete surgical resection remains the gold standard for primary or recurrent chondrosarcoma [2,3]. Due to poor radiosensitivity, high doses are recommended in palliative settings, after incomplete resection or for unresectable tumors in anatomically challenging sites. Particle therapy with protons or carbon ions (C-ions) provide enhanced local control and patients’ survival rates compared to conventional photon beam therapy [4].

The primary rationale for radiotherapy with particles is the dosimetric ballistics, i.e., the sharp dose increases at a well-defined depth (Bragg peak) and the rapid dose fall-off beyond that maximum [5]. Thus, a highly conformal high-dose region can be tailored to cover the target volume with high precision and to simultaneously reduce normal tissue exposure. C-ions have an elevated relative biological effectiveness (RBE) and are the current particle of choice for radiation-sensitive tumours. However, due the limited availability of dedicated C-ion treatments centers, their main clinical use is radioresistant skull base tumours, chordomas, and inoperable chondrosarcomas [6,7,8]. For this reason, there is a lack of large randomized, prospective studies as well as extensive basic cellular research despite clinical application in chondrosarcomas.

In our previous study, we were able to show that both X-ray and proton irradiation (IR) resulted in reduced chondrosarcoma cell survival and a G_2_/M phase arrest of the cell cycle. This was accompanied by a reprogramming in cellular metabolism. Interestingly, within 24 h the majority of clearly visible DNA damage after proton IR were repaired and the metabolic phenotype restored [9]. These highly efficient DNA repair mechanisms regenerate the majority of DNA damage and thus prevent a therapeutic success. Inhibiting this cellular response is therefore the logical consequence of improving radiosensitivity.

When DNA damage occurs, a complex network of signaling cascades is activated to ensure the survival of the cell. These tightly regulated processes stop the cell cycle, for example, to give the cell sufficient time to repair the lesion or trigger apoptosis/senescence [10]. Similar to other cellular signaling cascades, these repair mechanisms are activated by protein phosphorylation of three main upstream kinases—the Serine/Threonine kinase ATM, the Ataxia telangiectasia and Rad3-related kinase ATR, as well as the DNA-dependent protein kinase (DNA-PK) [11]. Another clinically interesting group of inhibitors are the Poly(ADP-ribose)-polymerase inhibitors (PARPis) [12]. Several PARPis were evaluated in a few sarcoma entities and provided promising data [13,14,15]. The capacity of the PARPi Olaparib to sensitize chondrosarcoma cell lines to radiotherapy was demonstrated [16].

In order to make a direct comparison of the promising PARPi with ATMi and ATRi, we evaluated the cell biological and radiosensitizing effects of the PARPis Olaparib and Veliparib in comparison to the ATMi Ku-55933 and the ATRi VE-821 on human chondrosarcoma cell lines, in association with conventional photon (X-ray) IR and high-LET particle therapy. We focused our interest on cell viability and proliferation, cell cycle distribution, the expression and phosphorylation of DNA damage markers, and the potential apoptotic induction following IR with and without the different types of DNA damage inhibitors to get an accurate vision of chondrosarcoma cell behavior following the combined treatment.

## 2. Results

### 2.1. Fractionation Experiments with X-ray IR Showed the Better Efficacy of the ATRi VE-821

After the one-hour pre-treatment with the PARPi Olaparib or Veliparib (Figure 1a), the ATMi Ku-55933, or the ATRi VE-821 (Figure 1b), the chondrosarcoma cells were irradiated in 3 groups each: 0 Gy (non-IR control), fractionated IR of 8 Gy X-ray on each of the three subsequent days, and a total dose of 24 Gy.

The dose-dependent reduction in viability is observed with all inhibitors, whereas VE-821 has the most efficient response. Large differences between the individual IR regimes could not be observed.

### 2.2. Olaparib and VE-821 Most Efficiently Prevented Regeneration of Chondrosarcoma Cells after Proton IR

One and twenty-four hours after pre-treatment with 10 µM inhibitors and 4 or 8 Gy proton IR, whole cell proteins from two different chondrosarcoma cell lines (SW-1353: Figure 2a,c; Cal78: Figure 2b,d) were isolated and the amount of double strand breaks was determined by quantifying γH2AX phosphorylation. Fold changes normalized to non-IR controls (Δ ratio; mean ± SD of *n* = 3) were presented in the Appendix A. Twenty-four hours after IR, the PARPi Olaparib and the ATRi VE-821 in particular continued to show strong phosphorylation, indicating permanent DNA damage in both cell lines. In addition, phosphorylation of the DNA damage markers ATM, BRCA, p53, and Rad51 was increased in a dose-dependent manner by proton IR, whereby Ku-55399 inhibited p-ATM and p-p53 most potently. Both PARPis, on the other hand, had hardly any influence on this regulation. These observations could be confirmed by γH2AX immunohistochemical staining. Figure 2e represents the quantification of the optical density (mean ± SD of *n* = 3, measured in biological triplicates).

Since the persistent effect on DNA damage of the ATRi VE-821 was the most promising, we focused on it in all further experiments that included also C-ions. Figure 3 shows the percentage depth dose curves of the proton and C-ion spread-out Bragg peak (SOBP) used for cell irradiations. For the flask position in the middle of the SOBP, the proton dose-averaged linear energy transfer (LET) value was 2.9 keV/μm (Figure 3a). The respective LET spectrum shows the increase with increasing depth in water. For C-ions in general, the LET is higher and the LET distribution is steeper, which is reflected by the scale of the mirrored *y*-axis (25 keV/μm versus 250 keV/μm). The respective LET range for C-ions with energies between 170 and 230 MeV/u was between 50 and 150 keV/µm, respectively. However, in the middle of the C-ion SOBP, the influence due to positioning errors was still below 0.5% with a LET of 55.2 keV/μm (Figure 3b).

### 2.3. The Additive Effect of Co-Treatment with the ATRi VE-821 on Cell Proliferation and Cell Cycle Distribution

Analysis of proliferation after co-treatment of VE-821 and X-ray/proton/C-ions IR of chondrosarcoma cells measured with the real-time xCELLigence system (OLS, Bremen, Germany) (Figure 4a) showed the additive effect of VE-821 and the respective type of IR.

Another important aspect in tumor biology is the impairment of the cell cycle by therapeutic interventions. Flow cytometry analysis was performed to determine the effect of different IR modes under the influence of VE-821 treatment on the cell cycle distribution of chondrosarcoma cultures when irradiated with 8 Gy X-rays/protons/C-ions. Non-IR cells and IR alone cells were measured as controls. All values from each of four individual experiments (% of gated cells) and their statistical differences are shown in Table 1 (mean ± SD, *n* = 4). Graphical representations of G_0_/G_1_, S, and G_2_/M values of both cell lines are shown in stacked bars (Figure 4b). High-LET IR with C-ions caused a highly significant increase in the number of cells in the G_2_/M phase compared to controls, which was accompanied by a decrease in the number of cells in the G_0_/G_1_ and S phases, indicating a sustained arrest in the G_2_/M phase at the 24 h time point. An additive VE-821 treatment significantly increased the number of gated cells in the G_0_/G_1_ phase and partly reversed the effect of IR. The Cal78 cell line responded with a significantly greater shift in cell cycle phases towards the S-phase, especially under the influence of VE-821 treatment. Representative flow cytometry measurements and the corresponding percentage values of the individual cell cycle phases of non-IR control cells (ctrl 0 Gy), VE-821-treated control cells (VE-821 0 Gy), C-ions-irradiated control cells (ctrl 8 Gy), and after combined treatment of VE-821 and 8 Gy C-ions are presented in Figure 4c.

### 2.4. Combined Treatment with VE-821 and Particle IR Increased MAPK Phosphorylation and Expression of Apoptosis Markers

Since protein phosphorylation is a fast-moving process, proteins for STAT3 and MAPK phosphorylation were isolated 1 h after combined treatment with 10 µM VE-821 ± 4 Gy X-ray/proton/C-ion IR. While STAT3 phosphorylation was significantly reduced by IR alone as well as by the combined treatment with VE-821, p-JNK and p-p38 increased specifically with the combined treatment (SW-1353: Figure 5a; Cal78: Figure 5b).

For the investigation of apoptotic induction, the corresponding proteins were isolated 24 h after IR. Here, too, a significant increase in caspases 8 and 9 expression and caspase 3 cleavage was observed, especially after the combined treatment with ATRi and IR (SW-1353: Figure 5c; Cal78: Figure 5d). All fold changes normalized to non-IR controls (Δ ratio; mean ± SD of *n* = 3) were presented in the Appendix A (SW-1353) and Appendix A (Cal78). The gene expression analysis of the cell survival marker survivin and the proliferation genes c-Myc and Ki67 is presented in Figure 5e. For all three genes, a highly significant reduction can be observed with additional treatment with VE-821.

### 2.5. Combined Treatment with VE-821 and Particle IR Resulted in Preservation of DNA Damage

To investigate to what extent the combined treatment influences the mechanisms of DNA repair, we isolated both whole cell lysates for western blotting and RNA for gene expression analysis 1 h, 24 h, and 72 h after 10 µM VE-821 ± 4 Gy X-ray/proton/C-ions. The number of double-strand breaks was determined by quantification of γH2AX phosphorylation and showed very clearly, especially after 24 h and 72 h, that the chondrosarcoma cells remain damaged by the combined treatment (SW-1353: Figure 6a; Cal78: Figure 6b). The expression of the DNA repair key players Rad51, MSH3, and XPC was reduced especially at the longer time points by the additional VE-821 treatment. The phosphorylation of the serine/threonine-specific checkpoint-kinase (CHK)1/2, BRCA1, and p53 were again most pronounced at the 1 h time point. The phosphorylation induced by X-ray/proton/C-ion IR was inhibited again by the supplementary treatment with VE-821. C-ions IR triggered the strongest responses. The results obtained were the same for both cell lines. All fold changes normalized to non-IR controls (Δ-ratio; mean ± SD of *n* = 3) were presented in the Appendix A (SW-1353) and Appendix A (Cal78).

In order to be able to show the regulation of the most important key genes after the combined treatment in more detail, we performed RT-qPCR analysis with RNA isolated 24 h after 10 µM VE-821 ± 8 Gy X-ray/proton/C-ions IR. The gene expression analysis of the DNA repair genes is presented in Figure 7. PT with proton or C-ions revealed a highly significant increase in XRCC1/2/3, XPC, and PCNA expression. MSH3/6, Rad51, and PRKDC were slightly inhibited in their mRNA expression or showed no changes. Light grey-striped bars represent SW-1353 and dark grey-dotted bars represent Cal78 chondrosarcoma cells. All DNA repair key genes revealed a highly significant reduced expression after the combined treatment with VE-821.

## 3. Discussion

The resistance of chondrosarcomas to conventional chemotherapy and radiotherapy significantly limits the therapeutic options. Sensitization of the tumor cells is therefore a key aspect in order to further develop and improve therapeutic approaches.

Particle therapy with protons and C-ions is a radiation treatment for radiosensitive tissue and/or radioresistant skull base tumours, chordomas, and inoperable chondrosarcomas [6]. However, their molecular mechanisms after IR are not well explored. When DNA damage occurs, a complex network of signaling cascades is activated to detect the DNA damage and stop the cell cycle. Cells then have enough time to repair the lesion or senescence/apoptosis occurs. Similar to other cellular signaling cascades, this process is driven by protein phosphorylation and at its center are three upstream kinases: ATM, ATR, and DNA-PKs [11]. Following the detection of DNA damage, these kinases ensure cell cycle arrest by phosphorylating and activating tumor suppressor checkpoint kinase 1 (CHK1), which regulates the G_2_/M checkpoint, preventing entry into mitosis [17]. In one of our previous papers, we were able to demonstrate that human chondrosarcoma cells have extraordinarily efficient DNA repair mechanisms after proton irradiation, regenerating most of the DNA damage caused by irradiation within 24 h and recovering the cell’s metabolism [9]. These repair mechanisms must be prevented in order to improve therapeutic intervention. For this reason, we applied a combined treatment with DNA repair inhibitors and particle irradiation.

A small Olaparib combination phase II open-label study in which patients with solid tumors harboring IDH1/2 mutations were treated with Olaparib as monotherapy revealed that three of five patients with chondrosarcomas had clinical benefit [18]. At the cellular and molecular biological level, hardly any studies have been published up to now. Olaparib decreased cell survival of CH2879 chondrosarcoma cells and the radiosensitivity was associated with mutations in homologous recombination repair genes, such as RAD50, SMARCA2, and NBN [16]. To inhibit the major kinase upstream pathways of DNA repair mechanisms, in this study we used the ATMi Ku-55933 and the ATRi VE-821 in addition to the PARPi Olaparib and Veliparib. Fractional application of X-ray IR showed the most efficient reduction of cell viability after a combined treatment with VE-821. Fractional use of particle IR on several consecutive days was not possible in our experimental setting due to limited beam time for preclinical research. The PARPi Olaparib and the ATRi VE-821 especially continued to show strong phosphorylation 24 h after proton IR, indicating permanent DNA damage in both cell lines. Since the persistent effect on DNA damage of the ATRi VE-821 was the most promising, we focused on it in all further experiments including C-ions.

The reproducible and accurate cell positioning in the IR measurement set-up is of high importance for studies investigating any biological effect in the gradient regions of the SOBP, namely the plateau and fall-off regions. The biological effect of the plateau region is of high clinical interest as it corresponds to the skin and healthy tissue of the patient before reaching the tumor target. In the fall-off region, where the dose and LET gradient are steepest, the particles deposit their whole (protons) or most (C-ions) of their energy abruptly over a small distance, making uncertainties caused by organ motion or daily set-up variability higher than for conventional radiotherapy. For this reason, the biological effect of the fall-off region is also of clinical importance, as it has the highest LET and RBE contribution of the whole SOBP.

The real-time proliferation assay showed the additive effect of VE-821 and the respective type of IR. While particle IR alone showed a significantly better effect than the conventional photon IR, the combined treatment with VE-821 reduced proliferation to a minimum. This cytotoxicity and increased radiosensitivity through VE-821 treatment has already been confirmed in other tumor types [19,20,21]. In a previous study, it was demonstrated that chondrosarcoma cells respond differently to IR due to their strong genetic heterogeneity [22]. Although our results of the two chondrosarcoma cell lines used were predominantly in agreement, we were also able to observe this heterogeneity in the cell cycle distribution after combined treatment. These differences are already evident in treatment with VE-821 without additional IR. While the untreated control cells of both cell lines were still in agreement, the VE-821 treatment caused a significantly higher number of cells to arrest in S-phase in the Cal78 cells than in the SW-1353 cells. This effect was then further enhanced by an additional X-ray/proton/C-ion IR. In SW-1353 cells, the high dose of 8 Gy resulted in a transformation in the cell cycle with a decrease in the number of cells in the G_0_/G_1_ phase and S phase, accompanied by a significant increase of the number of G_2_/M phase cells. While the cell cycle behavior is very similar after X-ray and proton IR, a highly significant arrest of the cells in the G_2_/M phase occurs after C-ion IR. Previously, Maity et al. showed that exposing a wide variety of cells to IR resulted in a mitotic delay that involved several events in G_0_/G_1_, G_2_/M, or S phase, and that the G_2_ arrest was observed in virtually all eukaryotic cells. The S phase delay was typically seen following higher doses (>5 Gy) [23].

Several signal transduction pathways stimulated by IR are mediated by the MAPK superfamily, including extracellular signal-regulated kinase (ERK), c-Jun N-terminal kinase (JNK), and p38 MAPK. JNK and p38 MAPK respond strongly to stress signals such as IR and chemotherapeutic agents. Activation of JNK and p38 MAPK by stress stimuli is closely associated with apoptotic cell death. It is also known that MAPK signaling may influence the radiosensitivity of tumor cells, as its activity is related to the radiation-induced DNA damage response [24]. For this reason, we analyzed the changes in MAPK phosphorylation and possible apoptotic induction after combined treatment with VE-821 and X-ray/proton/C-ion IR. While STAT3 phosphorylation is significantly reduced, both p-JNK and p-p38 increased with IR alone and especially with the combined therapy. Consistent with the MAPK phosphorylation results, there was increased expression of caspases 8 and 9 and caspase 3 cleavage, indicating apoptotic induction by the combined therapy.

It is also known that MAPK signaling can influence the radiosensitivity of tumor cells, as its activity is linked to the radiation-induced DNA damage response [24]. Photon IR mainly leads to isolated lesions such as single-strand breaks (SSBs), base damage, and double-strand breaks (DSBs). In contrast, particle therapy with high LET, such as C-ions, cause more localized and clustered DNA damage [25]. In a previous work, we were able to show that, in addition to homologous-directed repair (HDR) and non-homologous end joining (NHEJ), the mismatch-mediated repair (MMR) pathway with the main players EXO1, MSH3, and PCNA were clearly activated in chondrosarcoma cells after proton application [9]. This activation and thus upregulation of gene expression was particularly evident in XRCC1/2/3, XPC, and PCNA with all three types of IR, with C-ions eliciting the strongest response. The combined treatment with VE-821 caused a highly significant reduction in all investigated genes compared to the irradiated samples.

In summary, our data demonstrate that combined treatment with the ATMi VE-821 increases the radiation sensitivity of chondrosarcoma cells and significantly suppresses efficient DNA repair mechanisms in vitro. Thus, an important step has been taken towards improving the efficiency of radiotherapy.

## 4. Materials and Methods

### 4.1. Cell Culture

SW-1353 (primary grade II) (ATCC^®^ HTB-94™, LGC Standards, Middlesex, UK) and Cal78 (recurrence of dedifferentiated grade III) (ACC449; DSMZ, Leibniz, Germany) chondrosarcoma cell lines were cultured in Dulbecco’s-modified Eagle’s medium (DMEM-HG) supplemented with 10% FBS, 1% L-glutamine, 1% penicillin/streptomycin, and 0.25 µg amphotericin B (all GIBCO^®^, Invitrogen, Darmstadt, Germany). The cell lines were authenticated by STR profiling within the last three years. All experiments were performed with mycoplasma-free cells. For IR experiments, adherent chondrosarcoma cells in log-growth phase were plated either in a density of 1 × 10^5^ cells/Slideflasks 9 cm^2^ (Thermo Fisher Scientific) or 5 × 10^5^ cells/T25 flasks and incubated over night at 37 °C with 5% CO_2_.

### 4.2. Physical Parameters of Irradiation

All IR experiments were performed at MedAustron, the synchrotron-based Austrian center for ion therapy and research. The experimental research room is equipped with a horizontal beam line including an active spot scanning technique with active energy variation for both proton and C-ions. The precise and standardized positioning of samples embedded in respective measurement phantoms is facilitated by a high-precision robot couch and a laser positioning system. For the photon reference IR, a dedicated X-ray unit (YXLON Y.TU 320-D03, YXLON GmbH, Hamburg, Germany) was used. The unit is equipped with a 3 mm Be/3 mm Al/0.3 mm Cu filter. A current of 20 mA and a voltage of 200 kV were used to achieve a dose rate of 1.3 Gy/min. For the proton IR, a treatment plan with a spread-out Bragg peak (SOBP) of 4 cm was designed for a field size of 17 × 9 cm^2^ utilizing the treatment planning system (TPS) RayStation v7.99 (RaySearch Laboratories, Stockholm, Sweden). Dose calculation was performed with a Monte Carlo v4.3 dose engine [26]. The maximum proton energy was 124.7 MeV (range at 80% dose level in water = 101.2 mm).

Similarly, for the C-ion IR, a treatment plan with a spread-out Bragg peak (SOBP) of 4 cm was designed for the same field sizes, TPS, and dose engine. The maximum C-ion energy was 238.6 MeV/u (range at 80% dose level in water = 101.1 mm). Ripple filters were used to ensure a flat SOBP. For both particle types, the energy layers were spaced either 1 mm or 2 mm apart. The radiation delivery technique was pencil beam scanning, and no ranger shifters were used to modify the beam. The LET values for proton and C-ion IR were based on Monte Carlo calculations and were derived directly from the treatment planning system.

### 4.3. Viability and Proliferation Analysis

For the dose–response relationship, chondrosarcoma cells were pre-treated with 2.5 µM, 10 µM, and 25 µM of the PARPi Olaparib or Veliparib, the ATMi Ku-55933, or the ATRi VE-821 (all Selleckchem, Houston, TX, USA). Afterwards, the cells were irradiated with 0 Gy (non-IR control), with a fractionated IR of 8 Gy X-ray on each of the three subsequent days, or with a total dose of 24 Gy. Cell viability was determined with the CellTiter-Glo^®^ cell viability assay (Promega Corporation, Madison, MI, USA) and normalized to the non-IR controls. Background reference values were derived from the culture media. Absorbance was measured with a LUMIstar™ microplate luminometer (BMG Labtech GmbH, Ortenberg, Germany) (mean ± SD; *n* = 3, performed in biological quadruplicates).

The xCELLigence RTCA-DP device (OLS, Bremen, Germany) was used to monitor cell proliferation in real-time. Cells were seeded after IR in electronic microtiter plates (E-Plate™, OLS) and measured for 120 h according to the manufacturer’s instructions. Cell density was measured in quadruplicate with a programmed signal detection every 20 min. Data acquisition and analyses were performed with RTCA software (version 1.2, OLS).

### 4.4. Cell Cycle Analysis

Twenty-four hours after pre-treatment with 10 µM VE-821 and X-ray/proton/C-ions IR with 0 Gy (non-IR control), 4 Gy, and 8 Gy, cells were harvested by trypsinization and fixed with 70% ice-cold ethanol for 10 min at 4 °C. Before flow cytometry analysis, the cell pellet was resuspended in propidium iodide (PI)-staining buffer (50 μL/mL PI, RNAse A) and incubated for 15 min at 37 °C. Cell cycle distribution was measured with the CytoFlexLX (Beckman Coulter, Pasadena, CA, USA) and analyzed using the ModFit LT software Version 4.1.7 (Verity software house). Four independent experiments were conducted in each case.

### 4.5. Immunohistochemistry Staining

For immunohistochemical staining, cells were pre-treated with 10 µM Olaparib, Veliparib, Ku-55933, or VE-821 and irradiated with doses of 0 Gy (non-IR control) or 4 Gy proton. After 1 h and 24 h, cells were fixed with 4% paraformaldehyde for 30 min. Slides were incubated with γH2AX antibody (Merck, Darmstadt, Germany) for 1 h, the bridge antibody (Dako Agilent, Jena, Germany) for 30 min, the polymer (rabbit-ON-rodent-horse radish peroxidases; Biocare Medical, Pacheco, CA, USA) for 30 min, and AEC substrate chromogen (Dako) for 3 min. The reaction was stopped with PBS and then a hemalum core staining was performed. Pictures were taken with an Olympus BX51 microscope (Olympus, Vienna, Austria).

### 4.6. Protein Expression Analysis

Whole cell protein extracts were prepared with lysis buffer (50 mM Tris-HCl pH 7.4, 150 mM NaCl, 1 mM NaF, 1 mM EDTA, 1% NP-40, 1 mM Na3VO4) and a protease inhibitor cocktail (P8340; Sigma Aldrich, St. Louis, MO, USA) 1 h, 24 h, and 72 h after X-ray/proton/C-ions IR, respectively. Each protein was obtained with and without 10 µM VE-821 pre-treatment. Protein concentration was determined with the Pierce BCA Protein Assay Kit (Thermo Fisher Scientific). The proteins were separated by SDS-PAGE and were blotted on Amersham™ Protran™ Premium 0.45 µM nitrocellulose membranes (GE Healthcare Life Science, Little Chalfont, UK). Primary antibodies against the DNA damage key players phospho-histone γH2AX, Rad51, p-BRCA1, p-p53, p-ATM, MSH3, XPC; the serine/threonine kinases p-CHK1 and p-CHK2; the apoptotic proteases caspase 8, caspase 9, cleaved caspase 3; the apoptosis-inhibiting protein survivin (BIRC5), and β-actin (all Cell Signaling Technology, Danvers, MA, USA) as loading control were used. MAPK activity was measured with the following phospho-antibodies: p-STAT3, p-JNK, and p-p38. These were normalized to the respective unphosphorylated proteins (all Cell Signaling Technology). Blots were developed using a horseradish peroxidase-conjugated secondary antibody (Dako) for 1 h and the Amersham™ ECL™ prime western blotting detection reagent (GE Healthcare). Chemiluminescence signals were detected with the ChemiDocTouch Imaging System (BioRad Laboratories Inc., Hercules, CA, USA) and images were processed with the ImageLab 5.2 Software (BioRad Laboratories Inc.).

### 4.7. Reverse Transcription Polymerase Chain Reaction (RT-PCR)

Total RNA was isolated 24 h after IR with 4 Gy X-ray/proton/C-ions using the RNeasy Mini Kit and DNase-I treatment according to the manufacturer’s manual (Qiagen, Hilden, Germany). Two µg RNA were reverse transcribed with the iScript-cDNA Synthesis Kit (BioRad Laboratories Inc.) using a blend of oligo(dT) and hexamer random primers. Amplification was performed with the SsoAdvanced Universal SYBR Green Supermix (BioRad Laboratories Inc.) using technical triplicates and measured by the CFX96 Touch (BioRad Laboratories Inc.). The following QuantiTect primer assays (Qiagen) were used for real time RT-PCR: CCND1, c-Myc, XRCC1, XRCC2, XRCC3, XPC, MSH3, MSH6, PCNA, Rad51, and PRKDC. Results were analyzed using the CFX manager software for CFX Real-Time PCR Instruments (BioRad Laboratories Inc., version 3.1) and quantification cycle values (C_t_) were exported for statistical analysis. Results with Ct values greater than 32 were excluded from analysis. Relative quantification of expression levels was obtained by the ΔΔCt method based on the geometric mean of the internal controls ribosomal protein, large, P0 (RPL), and TATA box binding protein (TBP), respectively. Expression level (C_t_) of the target gene was normalized to the reference genes (ΔC_t_) and the ΔC_t_ of the test sample was normalized to the ΔC_t_ of the control (ΔΔC_t_). Finally, the expression ratio was calculated with the 2^−ΔΔCt^ method.

## Figures and Tables

**Figure 1 ijms-24-02315-f001:**
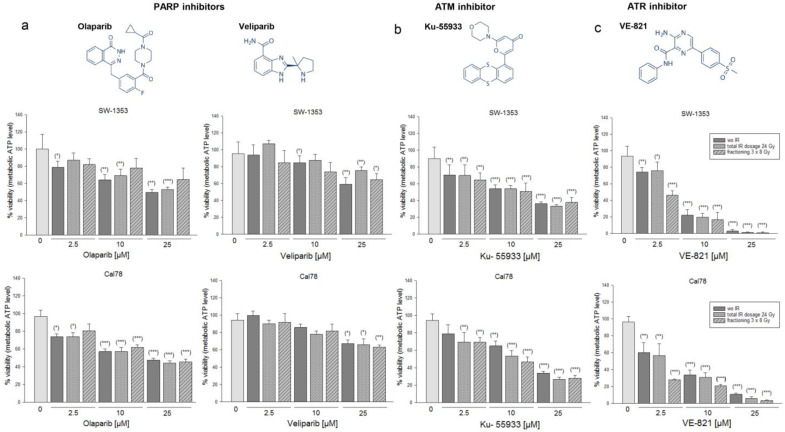
Fractionation experiments with reference X-ray IR under the influence of different types of DNA damage inhibitors. Chemical structures of (**a**) the PARPi Olaparib and Veliparib, (**b**) the ATMi Ku-55933, and (**c**) the ATRi VE-821. Cell growth of two chondrosarcoma cell lines SW-1353 and Cal78 was inhibited in a dose-dependent manner by combined treatment of 2.5, 10, or 25 µM inhibitor and X-ray IR (mean ± SD, *n* = 3, measured in biological quadruplicates). Statistical significances are defined as follows: * *p* < 0.05; ** *p* < 0.01; *** *p* < 0.001. IR regimes were 0 Gy (non-IR control; dark gray), fractionated IR of 8 Gy on three subsequent days (dashed), and the total dose of 24 Gy (dotted). Viability was measured 24 h after IR. VE-821 showed the most efficient response.

**Figure 2 ijms-24-02315-f002:**
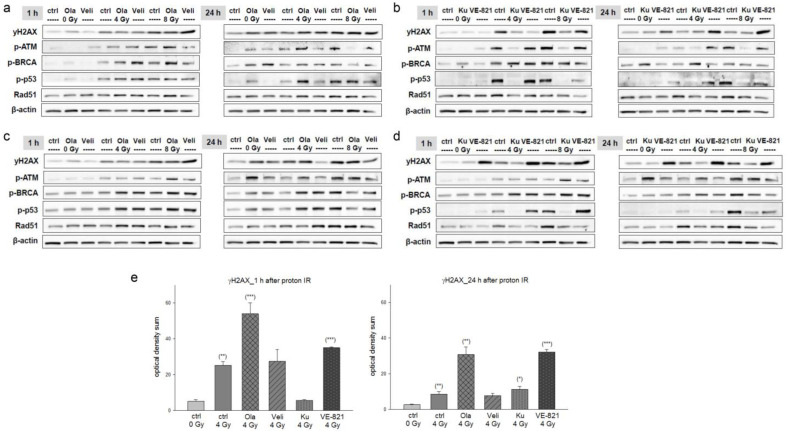
Olaparib and VE-821 most efficiently prevented regeneration of chondrosarcoma cells after proton IR. Protein phosphorylation pattern after combined treatment of 10 µM Olaparib (Ola) and Veliparib (Veli) and 0 Gy (unirradiated controls), 4 Gy, and 8 Gy proton IR of (**a**) SW-1353 and (**c**) Cal78 chondrosarcoma cells. Whole cell proteins were isolated 1 h and 24 h after IR. The corresponding co-treatments with Ku-55933 (Ku) and VE-821 are showed in (**b**) for SW-1353 and in (**d**) for Cal78 cells. Protein phosphorylation of the DNA damage marker γH2AX, ATM, BRCA, p53, and Rad51 was evaluated by immunoblotting. β-actin was used as loading control. Changes were presented as fold change (Δ-ratio) normalized to non-IR controls (mean ± SD of *n* = 3) (see Appendix A). (**e**) Immunohistochemical staining of γH2AX 1 h and 24 h after combined treatment of 10 µM inhibitors and 4 Gy proton IR (mean ± SD; *n* = 5; measured in triplicates). Non-IR cells were used as controls (ratio = 1). Statistical significances are defined as follows: * *p* < 0.05; ** *p* < 0.01; *** *p* < 0.001. Both at the protein level and in immunohistochemical staining, VE-821 showed the highest effects.

**Figure 3 ijms-24-02315-f003:**
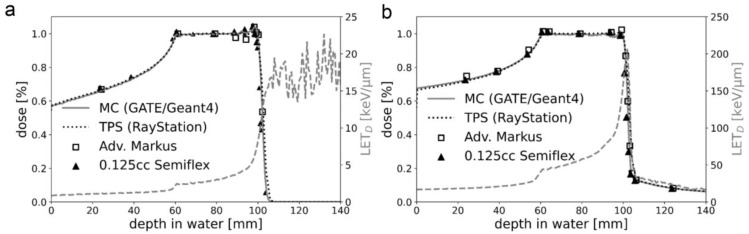
Physical characteristics of particle IR. The percentage depth dose curves of the (**a**) proton and (**b**) C-ion SOBP used for cell IR. A very good agreement between the predicted dose distributions of the treatment planning system (TPS) and the Monte Carlo (MC) simulation was achieved. The respective LET spectrum shows the increase with increasing depth in water. C-ions have a much higher LET than protons, which is reflected by the scale of the mirrored *y*-axis (25 keV/μm vs. 250 keV/μm). For protons, the LET after the fall-off region has no physical meaning as no particles are delivered beyond this depth.

**Figure 4 ijms-24-02315-f004:**
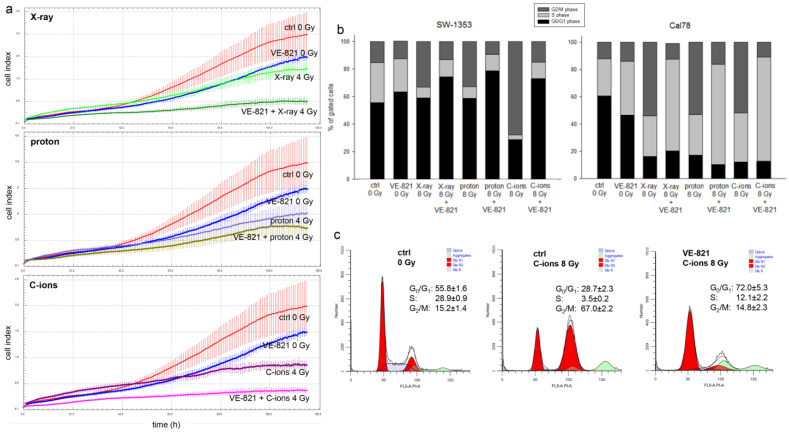
Proliferation and cell cycle distribution after the combined treatment of VE-821 and 4 Gy X-ray/proton/C-ions IR. (**a**) xCELLigence real-time proliferation analysis (red: non-IR controls; blue: VE-821 0 Gy; light green: X-ray 4 Gy; dark green: VE-821 + X-ray 4 Gy; purple: proton 4 Gy (LET 2.9 keV/μm); dark yellow: VE-821 + proton 4 Gy; dark magenta: C-ions 4 Gy (LET ca. 55 keV/μm); magenta: VE-821 + C-ions 4 Gy) shows the additive effect of the combined treatment, with the strongest effect with C-ions IR in SW-1353 cells. (**b**) Chondrosarcoma cells were analyzed using flow cytometry 24 h after combined treatment with 10 µM VE-821 and 8 Gy X-ray/proton/C-ions IR. The corresponding statistical evaluation is shown in stacked bar charts (*n* = 4). (**c**) Representative original tracks of non-IR control cells (ctrl), VE-821 0 Gy, and after 8 Gy C-ions IR without and with VE-821 treatment are shown.

**Figure 5 ijms-24-02315-f005:**
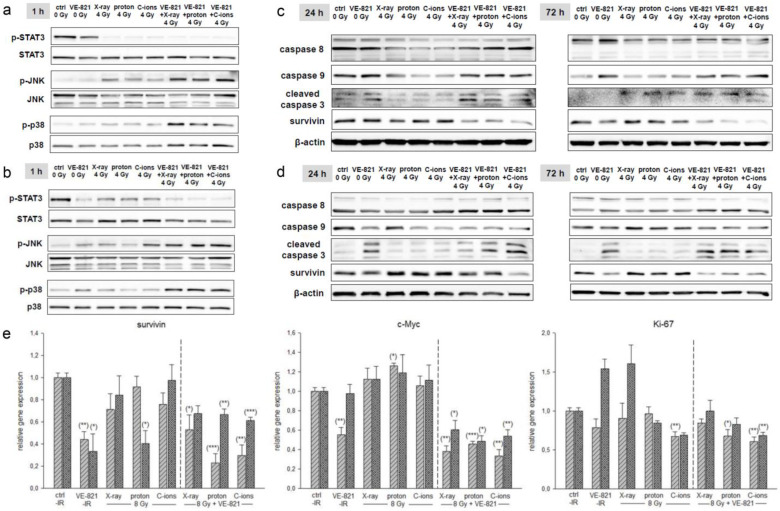
MAPK phosphorylation and the expression of apoptotic markers. p-STAT3, p-JNK, and p-p38 protein phosphorylation pattern of (**a**) SW-1353 and (**b**) Cal78 chondrosarcoma cells 1 h after the combined treatment with 10 µM VE-821 and 4 Gy X-ray/proton/C-ion IR. While STAT3 phosphorylation was significantly downregulated, JNK/p38 phosphorylation was increased by IR and further enhanced by VE-821. In addition, the combined treatment revealed a slight apoptotic induction in (**c**) SW-1353 and (**d**) Cal78. Changes were presented as fold change (Δ-ratio) normalized to non-IR controls (mean ± SD of *n* = 3) (see Appendix A). (**e**) Relative gene expression of the survival and proliferation markers survivin, c-Myc, and Ki67 24 h after combined treatment with 10 µM VE-821 and 8 Gy X-ray/proton/C-ions (SW-1353: light grey striped; Cal78: dark grey dotted) (mean ± SD; *n* = 6; measured in triplicates). Non-IR cells were used as controls (ratio = 1). Statistical significances are defined as follows: * *p* < 0.05; ** *p* < 0.01; *** *p* < 0.001.

**Figure 6 ijms-24-02315-f006:**
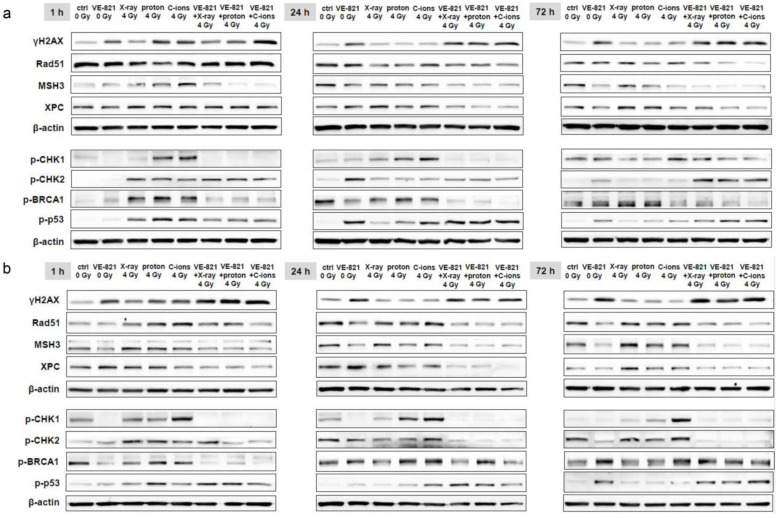
Protein phosphorylation pattern of DNA repair key players. The influence of the combined treatment with 10 µM VE-821 and 4 Gy X-ray/proton/C-ions IR on protein expression and phosphorylation of the DNA damage marker γH2AX, and of the DNA repair key players Rad51, MSH3, XPC, p-CHK1, p-CHK2, p-BRCA, and p-p53 were evaluated in (**a**) SW-1353 and (**b**) Cal78 cells by immunoblotting under non-IR control conditions (ctrl) after 1 h, 24 h, and 72 h. β-actin was used as loading control. Changes were presented as fold change (Δ-ratio) normalized to non-IR controls (mean ± SD of *n* = 3) (see Appendix A).

**Figure 7 ijms-24-02315-f007:**
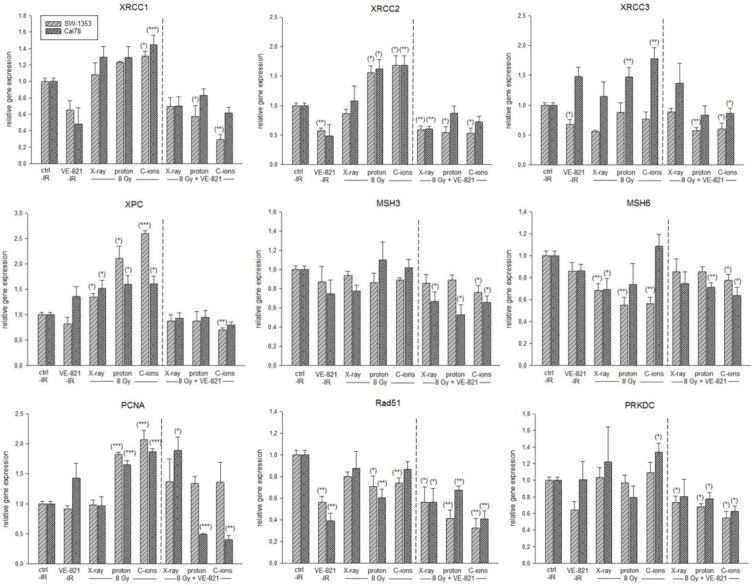
Relative gene expression of DNA repair key players. Relative gene expression of XRCC1/2/3, XPC, MSH3/6, PCNA, Rad51, and PRKDC 24 h after combined treatment with 10 µM VE-821 and 4 Gy X-ray/proton/C-ions IR in SW-1353 (light grey-striped) and Cal78 (dark grey-dotted) chondrosarcoma cells (mean ± SD; *n* = 6; measured in triplicates). Non-IR cells were used as controls (ratio = 1). Statistical significances are defined as follows: * *p* < 0.05; ** *p* < 0.01; *** *p* < 0.001.

**Table 1 ijms-24-02315-t001:** Cell cycle distribution of chondrosarcoma cells 24 h after combined treatment of 10 µM VE-821 and 8 Gy X-ray/proton/C-ions IR (*n* = 4; mean ± SD; n.s.: not significant; * *p* < 0.05, ** *p* < 0.01, *** *p* < 0.001 represent significances of the VE-821 effect; ## *p* < 0.01, ### *p* < 0.001 represent significances of irradiation).

	SW-1353			Cal78		
	G_1_/G_0_	S	G_2_/M	G_1_/G_0_	S	G_2_/M
ctrl 0 Gy	55.8 ± 1.6	28.9 ± 0.9	15.2 ± 1.4	60.4 ± 2.7	27.3 ± 3.2	12.1 ± 1.1
VE-821 0 Gy	63.4 ± 3.4 *	24.2 ± 1.0 ***	12.6 ± 2.7 n.s.	46.7 ± 5.9 *	39.1 ± 3.8 **	14.1 ± 2.1 n.s.
X-ray 8 Gy	59.0 ± 2.8	7.7 ± 1.1 ###	33.2 ± 2.2 ###	16.2 ± 1.8 ###	29.6 ± 2.5 n.s.	54.1 ± 1.5 ###
VE-821 + X-ray 8 Gy	74.4 ± 2.1 ***	12.6 ± 2.0 *	13.0 ± 0.3 ***	20.4 ± 3.3 *	67.2 ± 1.4 ***	11.6 ± 0.7 ***
proton 8 Gy	58.8 ± 3.0 n.s.	8.2 ± 1.3 ###	33 ± 1.8 ###	17.1 ± 1.1 ###	29.8 ± 2.3 n.s.	53.0 ± 1.2 ###
VE-821 + proton 8 Gy	78.7 ± 0.8 ***	11.8 ± 1.2 **	9.5 ± 1.8 ***	10.3 ± 0.8 ***	73.3 ± 2.1 ***	16.4 ± 1.4 ***
C-ions 8 Gy	28.8 ± 2.3 ###	3.6 ± 0.2 ###	67.7 ± 2.3 ###	12.1 ± 1.0 ###	35.9 ± 1.4 ##	52.0 ± 0.9 ###
VE-821 + C-ions 8 Gy	73.0 ± 5.2 ***	12.1 ± 6.2 *	14.8 ± 2.3 ***	12.7 ± 1.6 n.s.	76.4 ± 4.4 ***	10.9 ± 3.0 ***

## Data Availability

Not applicable.

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
