# Peer review of "The ATR Inhibitor VE-821 Enhances the Radiosensitivity and Suppresses DNA Repair Mechanisms of Human Chondrosarcoma Cells"

_ijms, 2023, doi:10.3390/ijms24032315_

Round 1

Reviewer 1 Report

Overcoming radioresistance is critical for treating cancer patients, more in sarcoma, particularly chondrosarcomas,  that are namely the more radio resistant cancer. Modify the dose of radiation, i.e. using hypofractionation, does not improve treatment whilst identifying new radio sensitizers seems to be right strategy.

Herein, the authors have investigated the combination of particles radiotherapy in combination with ATRi VE-821 in chondrosarcoma. The authors have done an excellent job analyzing every aspect that can characterize the efficacy of a radiosensitizer. I recommend evaluating the efficacy of the drug in vivo.

Author Response

Overcoming radioresistance is critical for treating cancer patients, more in sarcoma, particularly chondrosarcomas,  that are namely the more radio resistant cancer. Modify the dose of radiation, i.e. using hypofractionation, does not improve treatment whilst identifying new radio sensitizers seems to be right strategy.

Herein, the authors have investigated the combination of particles radiotherapy in combination with ATRi VE-821 in chondrosarcoma. The authors have done an excellent job analyzing every aspect that can characterize the efficacy of a radiosensitizer. I recommend evaluating the efficacy of the drug in vivo.

Authors reply: We are very pleased with the positive review of our work and thank Reviewer 1 for her/his time and efforts.

Regarding in vivo experiments, we can say that during the planning and preparation of the construction of MedAustron, the facility was intended for animal experiments, but it is not yet in operation. Unfortunately, we do not have the infrastructure to perform these experiments at the moment.

Reviewer 2 Report

The manuscript is well presented and the topic is interesting. I have just to main issues which I would like to see addressed:

1. the figures are so small that it is virtually impossible to read the captions and legends. 

2. I agree that the addition of VE-821 reduces viability and survival markers etc. However, my main problem is that I see very little effect of irradiation. The authors speak of an additive effect - and indeed, adding VE-821 is effective. But adding irradiation does not seem to change much. If significance was assessed with VE-821-treated, non-IR cells as a baseline, I am uncertain whether VE-821 IR-cells would really show a difference. This applies to almost all the results. In particular, Fig. 1 does not appear to show a difference between the irradiated and non-irradiated cells, which is actually in disagreement with figure 4 a (which cell line was used here? over what time was this followed? how do you explain the discrepancy?). Similarly for Figs. 5 and 7.

Author Response

The manuscript is well presented and the topic is interesting. I have just to main issues which I would like to see addressed:

Authors reply: Thank you for the positive feedback on our research. All comment have been carefully considered and were taken into account as explained in the detailed point-by-point reply below.

  1. the figures are so small that it is virtually impossible to read the captions and legends. 

Authors reply: The labeling of all figures has been enlarged as much as possible to make it more reader-friendly.

  1. I agree that the addition of VE-821 reduces viability and survival markers etc. However, my main problem is that I see very little effect of irradiation. The authors speak of an additive effect - and indeed, adding VE-821 is effective. But adding irradiation does not seem to change much. If significance was assessed with VE-821-treated, non-IR cells as a baseline, I am uncertain whether VE-821 IR-cells would really show a difference. This applies to almost all the results. In particular, Fig. 1 does not appear to show a difference between the irradiated and non-irradiated cells, which is actually in disagreement with figure 4 a (which cell line was used here? over what time was this followed? how do you explain the discrepancy?). Similarly for Figs. 5 and 7.

Authors reply: ad Fig.1: Since we can perform fractional irradiation on several consecutive days only with X-ray, particle IR data are missing in this experiment. At MedAustron, the particle accelerator is only available for research on weekends when it´s not needed for patient care.  The working groups are assigned 8 hours of particle irradiation time once a month.

As we have observed in previous studies, X-ray irradiation has little direct effect on chondrosarcoma cell viability.  However, the effect can be seen more clearly in the cell cycle distribution (Fig. 4b,c), for example. In the real-time proliferation assay (Fig. 4a), it can be seen that X-ray irradiation has less effect on the proliferation of chondrosarcoma cells than proton and C-ions irradiation. Furthermore, X-ray<proton<C-ion effects were detected at the level of protein phosphorylation (e.g., yH2AX, pCHK1, p-p53, pBRCA; Fig. 6 1 h samples) and at the level of gene expression of DNA damage markers (Fig. 7). The missing information on the cell line used has been added.

We thank Reviewer 2 for her/his time and efforts.

Reviewer 3 Report

Lohberger et al. presented an interesting topic of the combination between ATR inhibitor VE-821 and radiotherapy increased the treatment effects in human chondrosarcoma cells.

Further, they provided the evidence that enhanced therapeutic effect was boosted through suppressing DNA repairing pathway caused by radiation.

In general, this study is properly designed, and the results are clear and sound, and the references are relevant and updated.

1.     Please add statistical significance in Figure 1, particularly the IR-fraction groups.

2.     The study starts with X-ray, then switch to proton IR. Considering the differences between these two types of radiation methods, please justify. Also, in Figure 4, X-ray showed better combination effects with VE-821 compared to other two IR methods.

3.     In Figure 3, panel b showed rH2AX decreased at an order of control, Ku and VE-821; but in panel d, VE-821 was significantly higher than the other two samples. How to explain such inconsistency? Or data from a third cell line needed?

Author Response

Lohberger et al. presented an interesting topic of the combination between ATR inhibitor VE-821 and radiotherapy increased the treatment effects in human chondrosarcoma cells.

Further, they provided the evidence that enhanced therapeutic effect was boosted through suppressing DNA repairing pathway caused by radiation.

In general, this study is properly designed, and the results are clear and sound, and the references are relevant and updated.

Authors reply: Thank you for the positive feedback on our research. All comment have been carefully considered and were taken into account as explained in the detailed point-by-point reply below.

  1. Please add statistical significance in Figure 1, particularly the IR-fraction groups.

Authors reply: The corresponding siginificances were included in the figure. In addition, the labeling of all figures has been enlarged as much as possible to make it more reader-friendly.

  1. The study starts with X-ray, then switch to proton IR. Considering the differences between these two types of radiation methods, please justify. Also, in Figure 4, X-ray showed better combination effects with VE-821 compared to other two IR methods.

Authors reply: Compared to conventional radiotherapy with photons (X-ray), particle therapy is able to reduce the radiation exposure of adjacent healthy tissue and almost completely spare the tissue behind the tumor. Therefore, particle therapy is an ideal treatment for tumors near radiation-sensitive organs, such as the brain, spinal cord, eyes or lungs.

The advantage of particle therapy over conventional radiation therapy with photons is the completely different penetration behavior of the particles (particles) into the tissue. The interaction of the introduced particles with the tissue is strongly velocity-dependent. Thus, when passing through the tissue, the particles are continuously slowed down the deeper they penetrate. Only towards the end of their range do they develop their strong effect. There, the effect increases very strongly over a distance of a few millimeters, after which it drops to zero (for protons) or almost zero (for carbon ions). The deep dose profile produced in this process is called a Bragg peak. The energy of the particle as it leaves the accelerator determines the depth of penetration and the location of the maximum effect. This behavior makes it possible to deposit a very high radiation dose in the tumor while sparing other tissue located in front of the tumor - especially organs at risk.

In short version, this information is reflected in the introduction (lines 49-54).

Ad Figure 4: in the real-time proliferation assay (Fig. 4a), it can be seen that X-ray irradiation has less effect on the proliferation of chondrosarcoma cells than proton and C-ions irradiation. We have also shown this effect in other publications (e.g. Ref#9). This effect is demonstrated at the protein phosphorylation level (e.g. yH2AX, pCHK1, p-p53, pBRCA; Fig 6 1 h-samples) as well as at the gene expression level of the DNA damage markers (Fig. 7).

  1. In Figure 3, panel b showed rH2AX decreased at an order of control, Ku and VE-821; but in panel d, VE-821 was significantly higher than the other two samples. How to explain such inconsistency? Or data from a third cell line needed?

Authors reply: Thank you for checking our data so carefully. The discrepancy between the two cell lines in this case is only observed in the 1 h samples. As we have already shown in Lohberger et al, 2021 (Ref. #9), the cells are severely damaged 1 h after proton irradiation. However, the chondrosarcoma cells can regenerate within 24 h by highly efficient DNA repair mechanisms. This was also the starting point for the application of specific inhibitors. As can be seen very well in Fig. 2 b and d (24 h samples), after an additive treatment with VE-821 the cells STAY in a damaged state, which is essential for a therapeutic success.

We thank Reviewer 3 for her/his time and efforts.